# Metabolic syndrome and its associated factors among epileptic patients at Dessie Comprehensive Specialized Hospital, Northeast Ethiopia; a hospital-based comparative cross-sectional study

Altaseb Beyene Kassaw[1]*, Hiwot Tezera Endale[2], Kibur Hunie Tesfa[2], Meseret Derbew Molla[2]

1 Department of Biomedical Science, School of Medicine, College of Medicine and Health Sciences, Wollo University, Dessie, Ethiopia, 2 Department of Biomedical Science, School of Medicine, College of Medicine and Health Sciences, University of Gondar, Gondar, Ethiopia

* altasebbeyene7@gmail.com

## Abstract

### Introduction

Metabolic syndrome is a group of metabolic risk factors which are associated with an increased risk of cardiovascular disease and type2 diabetes. Nowadays, several studies have shown that the burden of metabolic syndrome is increasing among epileptic patients, and leads to MS-associated complications, including cardiovascular disease. However, getting published documents has been limited in Ethiopia and the study area. Therefore, we aimed to analyze the magnitude and associated factors of metabolic syndrome among epileptic patients in Dessie Comprehensive Specialized Hospital in compression with respective controls.

### Methods

Hospital-based comparative cross-sectional study design was implemented from June 25 to August 20, 2021. A total of 204 participants with an equal number of cases and controls (n = 102 each) were included. The data was collected through face-to-face interviews and biochemical analyses such as fasting blood glucose and lipid profiles were done through the enzymatic technique. The magnitude of metabolic syndrome was determined using both National Cholesterol Education Program Adult Treatment Panel III and International Diabetes Federation definition criteria. The STATA version 14 was used for statistical data analysis, and a comparison of categorical and continuous variables was done with $\chi^2$ and an independent t-test, respectively. The multivariable binary logistic regression analysis was used to identify factors associated with metabolic syndrome, and variables having a *P*-value of <0.05 were considered statistically significant.

**Data Availability Statement:** All relevant data are within the manuscript and its Supporting Information files.

**Funding:** The authors received no specific funding for this work.

**Competing interests:** The authors have declared that no competing interests exist.

**Abbreviations:** AEDs, Anti-Epileptic Drugs; BP, Blood Pressure; CBZ, Carbamazepine; CVDs, Cardio-Vascular Diseases; DBP, Diastolic Blood Pressure; DCSH, Dessie Comprehensive Specialized Hospital; FBS, Fasting Blood Sugar; GABA, Gamma-Aminobutyric Acid; HDL-C, High-Density Lipoprotein Cholesterol; IDF, International Diabetes Federation; LDL-C, Low-Density Lipoprotein Cholesterol; MS, Metabolic Syndrome; NCDs, Non-Communicable Diseases; NCEPATP III, National Cholesterol Education Program Adult Treatment Panel III; PWE, People with Epilepsy; T2DM, Type 2 Diabetes Mellitus; TC, Total Cholesterol; TG, Triacylglycerol; VPA, Valproic Acid; WC, Waist Circumference.

## Result

The prevalence of metabolic syndrome among the epileptic group was (25.5% in National Cholesterol Education Program Adult Treatment Panel III and 23.5% in International Diabetes Federation criteria), whereas it was 13.7% in National Cholesterol Education Program Adult Treatment Panel III and 14.7% in International Diabetes Federation criteria among control groups. According to the International Diabetes Federation criteria, low physical activity (adjusted odds ratio = 4.73, 95% CI: 1.08–20.68), taking multiple antiepileptic drugs (adjusted odds ratio = 8.08, 95% CI: 1.52–42.74), having a total cholesterol level of $\geq 200$ mg/dl (adjusted odds ratio = 5.81, 95%: 1.32–41.13) and body mass index (adjusted odds ratio = 1.57, 95% CI = 1.16–2.11) were significantly associated with metabolic syndrome among epileptic participants. Applying National Cholesterol Education Program Adult Treatment Panel III criteria, taking multiple antiepileptic drugs (adjusted odds ratio = 6.81, 95% CI: 1.29–35.92), having a total cholesterol level > 200 mg/dl (adjusted odds ratio = 7.37, 95% CI: 1.32–41.13) and body mass index (adjusted odds ratio = 1.53, 96% CI: 1.16–2.01) were also significantly associated.

## Conclusion

The prevalence of metabolic syndrome among epileptic patients was higher than that of control groups and reaches statistically significant by National Cholesterol Education Program Adult Treatment Panel III. Being on multiple antiepileptic drugs, body mass index, having low physical activity and raised total cholesterol were significantly associated with metabolic syndrome among the epileptic group. Therefore, it is better to focus on controlling weight, having sufficient physical exercise, and regular monitoring of total cholesterol levels in epileptic patients.

## Introduction

Epilepsy is a chronic neurological disease with two or more unprovoked seizures occurring more than 24 hours apart. It is characterized by recurrent seizures, which are brief episodes of involuntary movement as a result of excessive neural electrical discharges [1]. The epileptic population has a higher risk of non-communicable diseases (NCDs) such as CVDs, the notable culprit for the premature death of this group [2,3]. This is possibly due to the progressive emergence of atherosclerosis accelerating factors like obesity and the profound alterations of metabolic components, often called Metabolic syndrome (MS) [3]. A metabolic syndrome is a group of metabolic risk factors, including glucose intolerance, dyslipidemia, hypertension and central obesity which are associated with an increased risk of type 2 diabetes mellitus (T2DM), and CVDs [4]. It is a complex condition and originates primarily from an imbalance of calorie intake as well as energy expenditure but may also be affected by the genetic makeup of an individual, the predominance of a sedentary lifestyle and other factors like dietary patterns [5].

The syndrome has been progressively becoming one of the leading global public-health challenges and a threat to socio-economic developments owing to its association with increased risk of NCDs such as T2DM, atherosclerotic CVDs and all causes of profound morbidity as well mortality [6]. Globally, around 1/4 of the adult population is estimated to have MS even though it varies according to geographic area, age, race, sex and criteria used for

diagnosis [5]. They are twice as likely to die from it and three times more likely to have a heart attack or stroke compared with people without the syndrome.

Different studies have reported a high prevalence of MS in the epileptic population, ranging from 30.6% in a certain study [7] to 52.6% of another similar study [8]. It may be linked to seizure-related metabolic abnormalities, long-term antiepileptic medications use and a high occurrence of a sedentary lifestyle as well as other behavioral risk factors in the epileptic population [9,10]. Some antiepileptic medications, such as carbamazepine (CBZ) and phenytoin are known to alter the metabolic profile of an individual by altering the lipid profile, coagulation factors, and acute phase reactants [10]. The raised risk of MS in people with epilepsy (PWE) may also be due to a reduction in quality of life with reduced mental function and the presence of psycho-emotional stress [11]. Moreover, PWE has a significantly reduced life expectancy and a mortality rate two to three times higher compared with the general population, and it is estimated that CVDs account for 40 to 50% of this mortality [12].

In Ethiopia, the prevalence of MS differs across different areas and the pooled prevalence in the country of adult individuals was estimated to be (27.92%) [13]. However, its prevalence among PWE is not addressed yet, and to the scope of our understanding, there is a scarcity of studies documenting the magnitude as well as associated factors of MS among epileptic patients in the region since most of the studies have been conducted in Western countries, which could have several variations for other participants such as dietary, environmental, cultural and genetic differences with people living in Ethiopia. Considering the literature gap on MS prevalence and risk factors, this study aimed to assess the magnitude of MS and its associated factors among epileptic patients in comparison with apparently healthy control groups. The finding will help the physicians to consider the syndrome in the approaching epileptic patients and to provide the best appropriate care. Moreover, since most of the risk factors of MS are modifiable, prevention and control strategies have been suggested to prevent further complications and death for people with epilepsy. The results obtained from this study will also be served as baseline data for further studies and would help policymakers to have evidence for their action towards controlling MS among epileptic patients.

## Methods and materials

### Study setting, design and period

A hospital-based comparative cross-sectional study design was conducted from June 25-August 20/2021 at DCSH, Northeast Ethiopia. The DCSH is found in Dessie town of Amhara National Regional State, which is located 401 km northeast of the capital city (Addis Ababa) of Ethiopia. It serves as a referral center to South Wollo and surrounding Zones of about 7 million people including the neighboring Region. The hospital has divisions of units such as internal medicine, surgery, gynecology and obstetrics, pediatrics, oncology, psychiatry, laboratory, orthopedics, pharmacy and neurology. The neurology unit has an outpatient department, which registers and gives service to new, as well as follow-up patients, and an in-patient department. Around 1402 patients with epilepsy visited the outpatient department for follow-up in the eight months from July 2020 to February 2021. Accordingly, about 175 patients with epilepsy (on average) were visiting the hospital, in one month.

### Study participants

The source populations were all adult epileptic patients attending DCSH for cases and all adult patient attendants in the hospital for controls, whereas all adult epileptic patients attending DCSH during the data collection period for cases and all adult patient attendants present in the hospital during the data collection period for controls were the study population. To

minimize the effect of possible confounders, we chose controls (non-epileptic participants) from healthy patient attendants. Those patients whose ages were≥ 18 years old with diagnosed epilepsy (for cases), and apparently healthy participants who were ≥ 18 years old (for controls) were included in the study. Whereas, epileptic patients diagnosed with diabetes mellitus, hypertension and dyslipidemia predating the onset of epilepsy; other comorbid states like cancer, thyroid dysfunction, HIV; and clinically confirmed edematous as well as abdominally distended individuals, women who were at pregnancy or postpartum period of <6 months or on hormonal contraceptives; critically ill epileptic patients or with severe physical or mental disabilities were excluded from the study.

## Sample size and sampling procedures

The sample size was determined using Epi Info version 7, by taking 27.4% as the prevalence of raised blood pressure, a common component of MS, among controls from a previous study [14] and 50.0% as the expected prevalence among epileptic patients, since there was no previous study in Ethiopia. By taking the 1:1 case-to-control ratio, 95% CI, and 86% power, the total sample size becomes 186. When the non-response rate (10% = 18.6) was added, the final sample size became 204.6. So, 102.3≈ 102 epileptic patients and 102.3 ≈ 102 healthy participants were enrolled. To select participants from the study population, the monthly average epileptic patient flow was used. Epileptic study participants were chosen at regular intervals from their sequence of follow-up visits using systematic random sampling techniques. Equal number of age and sex-matched apparently healthy volunteer subjects were enrolled in the comparison group.

## Operational definitions

**Metabolic syndrome: was defined** based on the NCEP: ATP III criteria for the European population depending on the presence of at least three of the following parameters: abdominal obesity (WC>102 cm for males and >88 cm for females), raised BP (SBP ≥ 130 or DBP ≥85 mm Hg), TG ≥150 mg/dL, low HDL-C (<40 mg/dL for males and <50 mg/dL for females), raised FBS (≥ 110 mg/dL) [15]. It was also determined based on the IDF criteria; abdominal obesity (WC ≥94 cm for males and ≥ 80 cm for females) plus any two of the following four parameters: SBP ≥130 and/or DBP ≥ 85 mmHg or treatment of previously diagnosed hypertension, hypertriglyceridemia (≥150 mg/dL) or presence of treatment for this disorder, low HDL-C (<40 mg/dL for males and <50 mg/dL for females), or specific treatment for this lipid abnormality and raised FBS (≥100 mg/dL) or previously diagnosed T2DM [4].

   **Alcohol drinking status: alcohol drinker**; defined as the intake of any type of alcoholic beverage, such as beer, wine, or locally prepared alcoholic beverages for more than once per week in the past year regardless of the amount. **Non-drinkers**; are those who drink less than once per week for the last one year or never drink alcoholic products [16].

   **Khat chewing status: khat chewer;** participants who were chewing khat in any amount during the past one year, otherwise **non-chewer** [17].

   **Smoking status: smoker**; those who had a cigarette smoking practice within the last one year, whereas those participants who never smoked in their lifetime or smokers before one year were defined as **non-smokers** [17].

   **Fasting:** was defined as no caloric intake for at least 8 hours of the last meal [4].

   **Vigorous physical activity**: any activity that causes a large increase in breathing or heart rate if continued for at least 30 minutes (e.g. running, carrying, or lifting heavy loads, digging, or construction work) for a minimum of three days per week [18].

**Moderate-physical activity**: any activity that causes a small increase in breathing or heart rate (brisk walking or carrying light loads) that continued for at least 30 min with a minimum of 3 days per week or 5 or more days of these activities for at least 20 min per day or ≥3 days of vigorous-intensity activity per week of at least 20 min per day [18].

**Low-level physical activity**: A person not meeting any of the above-mentioned criteria for the moderate- or high-level categories [18].

**Low fruit and vegetable intake**: Consuming less than five servings (400 grams) of fruit and vegetables per day. For raw green leafy vegetables, 1 serving = one cup; for cooked or chopped vegetables, 1 serving = ½ cup; for fruit (apple, banana, orange, etc. . .), 1 serving = 1 medium size piece; for chopped, cooked and canned fruit, 1 serving = ½ cup; and for juice from fruit, 1 serving = ½ cup [18].

**Drug-responsive epilepsy**: Epilepsy in which the patient receiving the current AED regimen has been seizure-free for a minimum of 12 months or three times the maximum pretreatment interseizure interval, whichever is longer [19].

**Drug-resistant epilepsy**: Failure of adequate trials of two tolerated (a tolerable side effect profile) and appropriately chosen and used AED schedules (whether as monotherapies or in combination) to achieve sustained seizure freedom (for a minimum of 12 months or 3 times the maximum pretreatment interseizure interval, whichever is longer) [19]. An adequate trial is defined as continuous use of the AED for at least 3 months at a dose of at least 50% of the WHO's defined daily dose [20].

**Undefined drug responsiveness**: Drug responsiveness that cannot be classified as either drug-responsive or drug-resistant [19].

## Data and blood sample collection procedure

The data collection was conducted in accordance with the STEP-wise approach of the World Health Organization (WHO) for NCD surveillance in developing countries [18] and related pieces of literature. The approach had three levels: (1) interviewer-administered questionnaires were used to gather socio-demographic characteristics and information about lifestyle factors (it also contained a questionnaire related to epilepsy), (2) Anthropometric measurements (weight, height, BMI, waist circumference) and blood pressure were determined by using standardized devices/instruments (3) biochemical analyses were done to determine participants' Serum triglycerides (TGs), serum total cholesterol (TC), high-density lipoprotein cholesterol (HDL-C), and fasting blood glucose (FBG). As to prevent the spread of COVID-19, precautionary measures were applied throughout the whole procedure.

**Anthropometric measurements.** Physical measurements such as weight, height, waist circumference (WC) and BP were measured according to the WHO stepwise approach [18] and using adjusted equipment by two trained data collectors (nurses) who were working at DCSH neurologic clinic. The height was measured using a stadiometer. Data collectors instructed participants to stand upright, point feet outward; legs straight and knee together; arms at sides; head, shoulder blades, buttocks, and heels touching the measurement surface; look straight ahead and shoulder relaxed during the measurement. When the participants were weighed, they were asked to take off their shoes and other items that could add extra weight. Measurements for height and weight were approximated to the nearest 0.1 cm and 0.1 Kg, respectively. Then, BMI was calculated via weight in Kg divided by height in centimeter square (BM = $Kg/Ht^2$). Furthermore, WC was measured midway between the inferior angle of the ribs and the supra-iliac crest, with the erect stand-up position following normal out breathing by non-stretching tape to the nearest 0.1cm. Blood pressure was measured using a digital automatic BP monitor apparatus. When measuring it, the study subjects were instructed to sit

comfortably, with the back supported, legs uncrossed and the arm supported at heart level. After that, it was measured three consecutive times (with 5 minutes apart) after the participants had taken rest for at least 5 minutes or 30 minutes for those who take hot drinks like coffee, and the average values for both diastolic and systolic BP were considered for this study.

**Biochemical analysis.** For lipid profile and FBS analysis, approximately 5ml of blood sample was collected from the ante-capital vein through sterile technique after the study participants have overnight fasting or a minimum of 8 hours fasting and asked for their consent to give a sample. The blood sample was collected using an appropriate test tube. After the blood was clotted within 30 minutes, it was centrifuged at 3500 revolutions per minute for 5 minutes. A pure serum sample was separated, then placed in the neck tube and stored at -20˚c until processing. Fasting blood glucose, TC, HDL-C and TGs were determined using Dimension EXL 200 System chemistry analyzer through the enzymatic method from the serum sample, whereas LDL-C was calculated using the Freidwald formula: Total cholesterol (TC = HDL + LDL + TG/5, LDL-c = TC-HDL-TG/5 [21].

## Data processing and analysis

Data were checked for completeness then coded, entered and cleaned using Epi-Data version 4.6 statistical software, and exported to STATA version 14 for analysis. Descriptive analysis was carried out and results were presented using tables and figures. The categorical variables were explored using frequency and percentage, and the continuous variables were expressed as mean ± standard deviation. The chi-square ($\chi^2$) tests were used to compare categorical variables while continuous variables were compared using independent t-tests. The associations between MS and associated factors were investigated using the bivariable and multivariable binary logistic regression model. Variable with p-value < 0.25 in the bivariable logistic regression was fitted into the multivariable binary logistic regression model for final analysis. An adjusted odds ratio (AOR) with a 95% confidence level (CL) was used for the interpretation of the strength of prediction of the independent variables to the outcome (MS). Variables having a *P*-value of < 0.05 at 95% CL with multivariable logistic regression were considered statistically significant. The goodness of fit of the final logistic model was tested by using Hosmer and Lemeshow's test.

## Data quality control

To assure the quality of data, a pre-test was done before running the actual data collection to check completeness, consistency, sensitivity, and applicability and then modified accordingly. It was conducted by 5% of each group (total participants = 10) of the sample size of volunteer participants at a nearby hospital (Boru Meda General Hospital). The questionnaire was prepared in English and then translated to Amharic and back-re-translated to English to see its consistency. A one-day training was given by the principal investigator for data collectors regarding the objective of the study, methodology and relevance of the study before running the concrete data collection. The laboratory procedures were assured by strictly following the manufacturers' instructions and standard operational procedure (SOP) in the pre-analytic, analytic and post-analytic stages of laboratory service. After completion of the data collection, each questionnaire was checked for completeness, clarity and consistency on a daily basis. Data entry was also double-checked.

## Ethical consideration

The study was conducted following the ethical principles of the Declaration of Helsinki. To conduct the study, ethical issues were considered. The ethical clearance was obtained from the

Ethical Review Board of the University of Gondar, School of Medicine (reference number: 685/6/2021). An official permission letter was also granted from the managers of DCSH. Written informed consent was obtained from each participant with respect to their willingness before participating in the study. All the principles of ethics such as confidentiality and privacy were ensured throughout the study process. The study participants were informed that refusal to consent or withdrawal from the study does not negatively affect their access to health care.

# Result

## Sociodemographic characteristics of the study participants

A total of 204 study participants grouped into two age and sex-matched groups (102 epileptic and 102 healthy controls) with a response rate of 100% were enrolled. The mean (± SD) age of the participants for both groups was 34.5 (±11.62) years and 52 (51%) of each group were males. Nearly one-third, 32 (31.4%), of epileptic participants were unable to read or write and 36 (35.3%) of the controls have attended primary school. Among the total participants, 31 (30.4%) of the epilepsy group and 28 (27.5%) of the controls were farmers. More than half of the study participants, both in the epileptic group (64 (62.7%)) and control (66 (64.7%)), were urban residents. The mean (± SD) monthly income of the participants in the epileptic and control groups was 4603.92 (± 2498.95) and 5268.14 (± 2538.201) Ethiopian birr, respectively (Table 1).

## Behavioral characteristics of the study participants

Most of the study participants, 98 (91.1%) in the epileptic and 91 (89.2%) in the control group, had no history of smoking. The majority of the study participants in both groups, 91 (90.2%)

**Table 1. Socio-demographic characteristics of the study participants in DCSH, Dessie, Northeast Ethiopia, 2021 (n = 204).**

| Variables | Category | Epilepsy group (n = 102) [N (%)] | Control group (n = 102) [N (%)] |
|---|---|---|---|
| Age (mean ± SD) | N/A | 34.45 ±11.62 | 34.49 ±11.62 |
| Age category | 18–28 | 34 (33.3) | 36 (35.3) |
| | 29–39 | 36 (35.3) | 35 (34.3) |
| | ≥ 40 | 32 (31.4) | 31 (30.4) |
| Sex | Male | 52 (51) | 52 (51) |
| | Female | 50 (49 | 50 (49) |
| Occupation | Merchant | 13 (12.7) | 21 (20.6) |
| | Farmer | 31 (30.4) | 28 (27.5) |
| | Employee | 18 (17.6) | 23 (22.5) |
| | Daily labourer | 24 (23.5) | 16 (15.7) |
| | cOther | 16 (15.7) | 14 (13.7) |
| Educational status | unable to read/write | 32 (31.4) | 21 (20.6) |
| | Primary | 29 (28.4) | 36 (35.3) |
| | Secondary | 18 (17.6) | 19 (18.6) |
| | College/above | 23 (22.5) | 26 (25.5) |
| Residence | Urban | 64 (62.7) | 66 (64.7) |
| | Rural | 38 (37.3) | 36 (35.3) |
| Income (mean ± SD) | N/A | 4603.92± 2498.95 | 5268.14±2538.2 |

**Note**: cstudent and housewife.

**Abbreviations**: ETB = Ethiopian Birr, SD = standard deviation, N/A = not applicable.

in epileptic and 81 (80.4%) in control groups, were khat non-chewer at the time of data collection. Likewise, the highest number of study participants in both groups, 97 (95.1%) epileptics and 79 (77.5%) of the controls, were not reporting a history of alcohol drinking. A total of 39 (38.2%) participants in the epileptic group and 45 (44.1%) participants in the control group engaged in moderate physical activity. In addition, 63 (61.8%) of epileptic and 52 (51.0%) of the control respondents were consuming less than five servings of fruit and vegetables per day (Table 2).

## Clinical characteristics and therapy of the epileptic participants

The majority of the epileptic participants, 65 (63.7%), had generalized onset type of epilepsy and 91 (89.1%) of patients were on AEDs at the time of recruitment. Nearly two-thirds of the participants, 69 (67.6%), were on monotherapy in which phenobarbitone was the most frequently utilized drug, in 39 (38.2%) patients. The mean duration since epilepsy diagnosis was 5.9 (± 3.7) years, and 63 (61.8%) of the participants were responsive to anti-epileptic agents. Out of the total epileptic patients, only two participants were hypertensive with anti-hypertension medication, which was started after the onset of epilepsy and none of the study participants reported either previously T2DM or on treatment for abnormal TG and HDL-C, (Table 3).

## Anthropometric and biochemical parameters of the study participants

In this study, the average SBP levels of epilepsy and control groups were found to be 114.22 (± 11.75) and 116.91 (±13.16) mmHg, respectively. The mean ± SD levels of BMI for epilepsy and control groups were 22.37 ± 3.06 and 22.12 ± 2.38 kg/m$^2$, respectively. Besides, the results of the present study showed that in the serum of epilepsy and control participants, the average HDL-c levels were 45.68 ± 9.01 and 48.21± 8.43 mg/dl, respectively. A significant difference was found in the mean of HDL-c, WC and FBS levels between epilepsy and control groups with respective P-values of 0.040, 0.002 and 0.006 (Table 4).

**Table 2. Behavioral characteristics of study participants in DCSH, Dessie, Northeast Ethiopia, 2021 (n = 204).**

| Variables | Category | Epilepsy group (n = 102) [N (%)] | Control group (n = 102) [N (%)] |
|---|---|---|---|
| Physical activity level | Vigorous | 25 (24.5) | 35 (34.3) |
| | Moderate | 39 (38.2) | 45 (44.1) |
| | Low | 38 (37.3) | 22 (21.6) |
| Days of FEV intake/ Week | ≥5 | 16 (15.7) | 26 (25.5) |
| | 3–5 | 34 (33.3) | 36 (35.3) |
| | <3 | 52 (51.0) | 40 (39.2) |
| Number of servings of FEV on average / day | <5servings | 63 (61.8) | 52 (51.0) |
| | ≥5 servings | 39 (38.2) | 50 (48.0) |
| Khat chewing status | Chewer | 10 (9.8) | 20 (19.6) |
| | Non-chewer | 91 (90.2) | 81 (80.4) |
| Alcohol consumption | Drinker | 5 (4.9) | 23 (22.5) |
| | Non-drinker | 97 (95.1) | 79 (77.5) |
| Cigarette smoking | Smoker | 4 (3.9) | 11 (10.8) |
| | Non-smoker | 98 (96.1) | 91 (89.2) |

**Abbreviation**: FEV = fruit and/or vegetable.

**Table 3. Clinical characteristics and therapy of epileptic group at DCSH, Dessie, Northeast Ethiopia, 2021 (n = 102).**

| Variables | Category | Frequency | (%) |
|---|---|---|---|
| Epilepsy subtype | Generalized onset | 65 | 63.7 |
| | Focal onset | 15 | 14.7 |
| | Unknown Onset | 22 | 21.6 |
| Duration of epilepsy, years (mean ± SD) | N/A | 5.9 ± 3.7 | N/A |
| Currently on anti-epileptic treatment | Yes | 91 | 89.1 |
| | No | 11 | 10.9 |
| Current AEDs combination | 0 (Currently not on AEDs | 11 | 10.8 |
| | 1 (On monotherapy) | 69 | 67.6 |
| | ≥ 2 (On Poly therapy) | 22 | 21.6 |
| Name of Current AEDs combination | Currently not on medication | 11 | 10.8 |
| | Phenobarbitone | 39 | 38.2 |
| | Phenytoin | 22 | 21.6 |
| | Carbamazepine | 4 | 3.9 |
| | Phenobarbitone and Phenytoin | 16 | 15.7 |
| | Phenobarbitone and Carbamazepine | 8 | 7.8 |
| | Phenobarbitone and valproic acid | 1 | 1.0 |
| | Phenobarbitone, Phenytoin and Carbamazepine | 1 | 1.0 |
| Duration since AEDs started, years (mean ± SD) | N/A | 4.6 ± 3.5 | N/A |
| Drug responsiveness status | Drug responsive | 63 | 61.8 |
| | Drug-resistant | 10 | 9.8 |
| | Undefined | 29 | 28.4 |
| Previously T2DM or currently taking medication | Yes | 0 | 0 |
| | No | 102 | 100 |
| Known hypertensive /on medication | Yes | 2 | 2.0 |
| | No | 100 | 98.0 |
| On treatment for lipid abnormalities | Yes | 0 | 0 |
| | No | 102 | 100 |

**Table 4. Levels of anthropometric and biochemical parameters of the study participants at DCSH, Dessie, Northeast Ethiopia, 2021 (n = 204).**

| Variables | Epilepsy group (n = 102) | Control group (n = 102) | [a]P-value |
|---|---|---|---|
| | Mean ± SD | Mean ± SD | |
| SBP | 114.22 ± 11.75 | 116.91 ± 13.16 | 0.124 |
| DBP | 76.50 ± 7.91 | 77.08 ± 7.81 | 0.600 |
| HDL-c | 45.68 ± 9.01 | 48.21 ± 8.43 | *0.040 |
| BMI | 22.37± 3.06 | 22.12 ± 2.38 | 0.510 |
| WC | 81.79 ± 9.17 | 77.75 ± 8.95 | *0.002 |
| FBS | 90.37 ± 16.17 | 84.75 ± 12.56 | *0.006 |
| TC | 178.45± 41.79 | 177.91 ± 39.22 | 0.924 |
| LDL-c | 107.81 ± 35.48 | 105.31 ± 34.17 | 0.610 |
| TG | 125.09 ±56.41 | 126.44 ± 53.37 | 0.816 |

**Note**: *Statistically significant difference,

[a]P- value was derived from the independent t-test.

**Abbreviations**:—SBB = systolic blood pressure, DBP = diastolic blood pressure, HDL-c = high-density lipoprotein cholesterol, BMI = body mass index, WC = waist circumference, FBS = fasting blood sugar, TC = total cholesterol, LDL-c = low-density lipoprotein cholesterol, TG = triglyceride.

## The magnitude of metabolic syndrome and its components among the study participants

According to the NCEP-ATP III criteria, the prevalence rate of MS was found to be 25.5% (95% CI: 17.03% - 33.95%) among the epileptic group and 13.7% (95% CI: 7.04%- 20.40%) among the control group. Likewise, the prevalence rate was estimated based on IDF definition and found to be 23.5% (95% CI: 15.28% - 31.76%) among epileptic participants and 14.7% (95% CI: 7.83% - 21.58%) among the controls. The difference reaches statistical significance for the NCEP-ATP- III definition (p = 0.034), however, there was no statistically significant difference for IDF (p = 0.109) (Table 5). According to both criteria, the prevalence rate of MS varied in each age category of the two study groups' participants, and a high prevalence rate was found in the age group of ≥ 40 years in both groups (34.5% among epilepsy and 22.6% among control using both criteria). The study also showed that a different prevalence of MS between males and females in the two groups was observed with a high prevalence rate (30%) among epilepsy female participants based on the NCEP-ATP III definition (Fig 1).

Regarding the prevalence of components of MS, reduced HDL-C was the most prominent in both groups, at 40.2% in epileptic participants and 26.5% in controls, followed by elevated TG levels with the prevalence rate of 33.3% and 20.6% for epilepsy and control groups, respectively (Table 5).

**Table 5. Frequency of metabolic syndrome and its components among the study groups and the related comparisons at DCSH, Dessie, Northeast Ethiopia, 2021.**

| Components | Epilepsy group N (%) | Control group N (%) | [a]P-value |
|---|---|---|---|
| Central obesity by NCEP-ATPIII | | | 0.286 |
| Yes | 15 (14.7) | 10 (9.8) | |
| No | 87 (85.3) | 92 (90.2) | |
| Central obesity by IDF | | | 0.187 |
| Yes | 28 (27.5) | 20 (19.6) | |
| No | 74 (72.5) | 82 (80.4) | |
| HDL-c level | | | *0.038 |
| Low | 41 (40.2) | 27 (26.5) | |
| Normal | 61 (59.8) | 75 (73.5) | |
| TG level | | | *0.040 |
| Increased | 34 (33.3) | 21 (20.6) | |
| Normal | 68 (66.7) | 81(79.4) | |
| BP | | | 0.713 |
| Elevated | 17 (16.7) | 19 (18.6) | |
| Normal | 85 (83.3) | 83 (81.4) | |
| FBG by NCEP-ATP III | | | *0.005 |
| Increased | 16 (15.7) | 4 (3.9) | |
| Normal | 86 (84.3) | 98 (96.1) | |
| FBG by IDF | | | *0.034 |
| Increased | 26 (25.5) | 14 (13.7) | |
| Normal | 76 (74.5) | 88 (86.3) | |
| MS- according to NCEP-ATP III criteria | | | *0.034 |
| Yes | 26 (25.5) | 14 (13.7) | |
| Non | 76 (74.5) | 88 (86.3) | |
| MS- according to IDF criteria | | | 0.109 |
| Yes | 24 (23.5) | 15 (14.7) | |
| No | 78 (76.5) | 87 (85.3) | |

**Note**: *Statistically significant difference,
[a] P-value from Chi-square test.

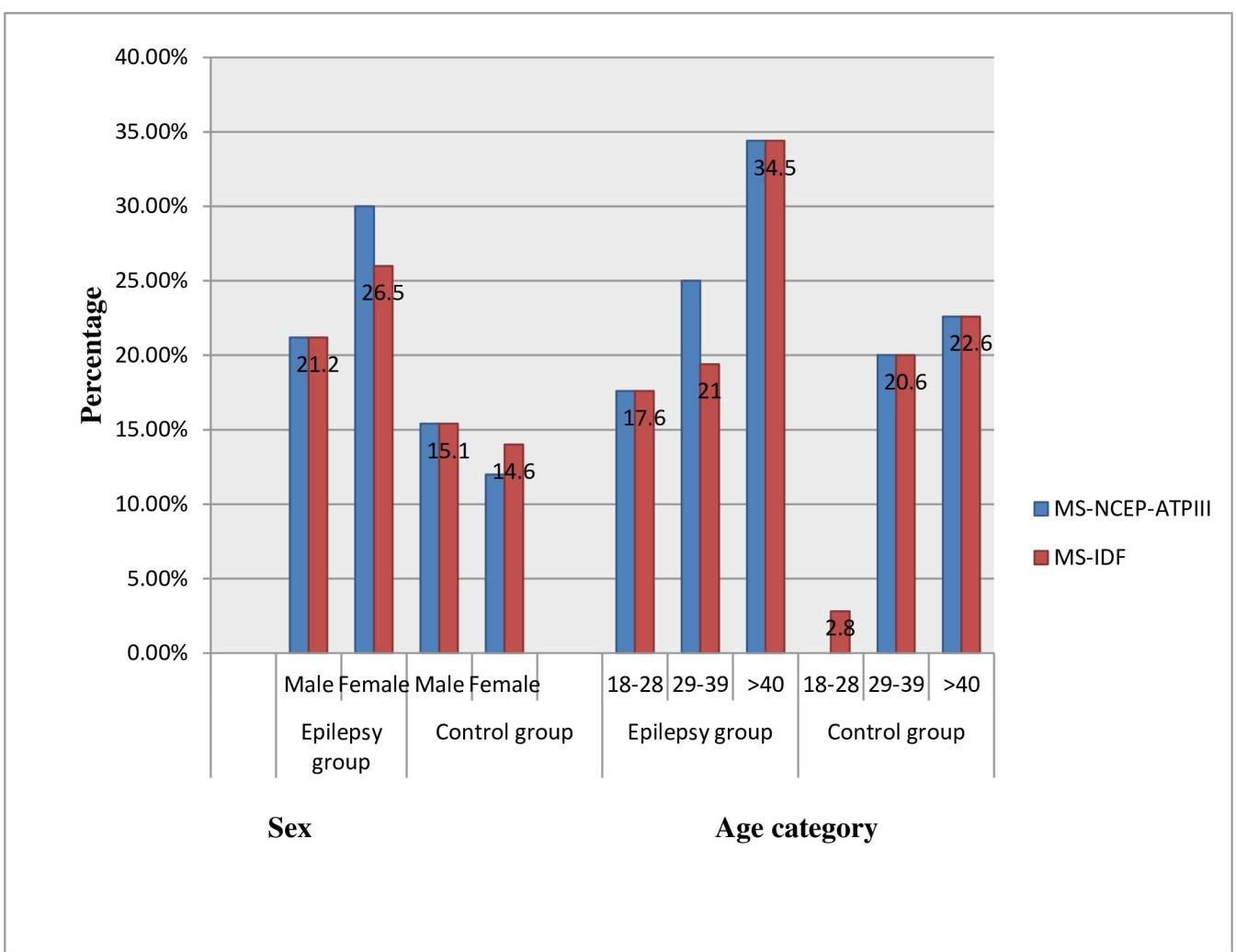

**Fig 1. The magnitude of metabolic syndrome among study groups stratified by sex and age group (according to ATP-III and IDF definitions).**

## Associated factors of metabolic syndrome among the epileptic groups

In bivariable analysis, age, residence, level of physical activity, FEV intakes status, epilepsy subtype, epilepsy duration, current AEDs use, drug responsiveness status, BMI, TC and LDL-C had P-value < 0.25 and were fitted into the multivariable binary logistic regression model for final analysis, by both IDF and ATP-III criteria.

Findings of a multivariable analysis with IDF criteria showed that having a low level of physical activity, being on multiple AEDs, BMI, and having TC ≥ 200 mg/dl were found to have significant associations with MS (P-value < 0.05). Participants with low physical activity had a 4.7 (AOR = 4.73, 95% CI: 1.08–20.68) times higher risk of having MS compared to sufficient activity. The odds of having MS among participants taking multiple anti-epileptic medications was 8.1 (AOR = 8.08, 95% CI: 1.52–42.74) times higher compared to participants who were taking a single medication. Likewise, participants who had a TC level of ≥ 200 mg/dl were 5.8 (AOR = 5.81, 95%: 1.32–41.13) times more likely to have MS compared to patients who had < 200 mg/dl. For a unit increase in BMI of a participant, the odds of having MS increases by a factor of 1.6 (AOR = 1.57, 95% CI = 1.16–2.11) (Table 6). Similarly, applying the

**Table 6. Bivariable and multivariable binary logistic regression analysis of factors associated with metabolic syndrome among the epileptic group using IDF criteria at DCSH, Dessie, Northeast Ethiopia, 2021(n = 102).**

| Variables | Categories | MS- IDF | | COR (95% CI) | AOR (95% CI) | p-value |
|---|---|---|---|---|---|---|
| | | Yes (n (%)) | No (n (%)) | | | |
| Age category | 18–28 | 6 (17.6) | 28 (82.4) | 1 | 1 | |
| | 29–39 | 7 (19.4) | 29 (80.6) | 1.13 (0.34–3.77) | 1.15 (0.18–7.19) | 0.883 |
| | ≥40 | 11 (34.4) | 21 (65.6) | 2.44 (0.79–7.68) | 2.39 (0.32–18.02) | 0.396 |
| Residence | Rural | 13 (34.2) | 25 (65.8) | 1 | 1 | |
| | Urban | 11 (17.2) | 53 (82.8) | 0.39 (0.17–1.01 | 0.34 (0.07–1.64 | 0.177 |
| Level of physical activity | Sufficient | 9 (13.8) | 56 (86.2) | 1 | 1 | |
| | Low | 15 (40.5) | 22 (59.5) | 4.24(1.62–11.11) | 4.73 (1.08–20.68) | *0.039 |
| Level of FEV intakes | Adequate | 6 (13.3) | 39 (86.7) | 1 | 1 | |
| | Low | 18 (31.6) | 39 (68.4) | 3.00 (1.08–8.36) | 3.80 (0.79–18.36) | 0.096 |
| Epilepsy subtype | Generalized onset | 17 (26.2) | 48 (73.8) | 1 | 1 | |
| | Focal onset | 1 (6.7) | 14 (93.3) | 0.2 (0.02–1.65) | 0.86 (0.06–12.61) | 0.915 |
| | Unknown Onset | 6 (27.3) | 16 (72.7) | 1.06 (0.36–3.15) | 1.32 (0.24–7.31) | 0.752 |
| Epilepsy dura-tion (mean±SD) | N/A | 7.6 ± 4.7 | 5.4 ± 3.2 | 1.7 (1.03–1.32) | 1.01 (0.83–1.23) | 0.930 |
| Current AEDs use | On monotherapy | 11 (6.2) | 57 (83.8) | 1 | 1 | |
| | On Poly therapy | 12 (52.2) | 11 (47.8) | 5.65(1.99–12.03) | 8.08 (1.52–42.74) | *0.014 |
| | Not on AE-agents | 1 (9.1) | 10 (90.9) | 0.52 (0.06–4.47) | 1.95 (0.14–27.19) | 0.621 |
| Drug responsiveness status | Drug responsive | 18 (28.6) | 45 (71.4) | 1 | 1 | |
| | Drug-resistant | 2 (20.0) | 8 (80.0) | 0.66 (0.13–3.43) | 0.15 (0.01–2.37) | 0.180 |
| | Undefined | 4 (13.8) | 25 (86.2) | 0.53 (0.17–1.61) | 0.44 (0.708–2.48) | 0.351 |
| BMI (mean ±SD) | N/A | 24.8±2.9 | 21.6±2.7 | 1.46 (1.20–1.77) | 1.57 (1.16–2.11) | *0.003 |
| TC | <200 | 13 (17.6) | 61 (82.4) | 1 | 1 | |
| | ≥ 200 | 11 (39.3) | 17 (60.7) | 3.04 (1.16–7.98) | 5.81 (1.03–32.91) | *0.047 |
| LDL-C | <130 | 12 (16.2) | 62 (83.8) | 1 | 1 | |
| | ≥ 130 | 12 (42.9) | 16 (57.1) | 3.88 (1.47–10.23 | 0.78 (0.16–3.86) | 0.762 |

**Note**: *Statistically significant factors associated with MS, 1 = Reference.

**Abbreviations**: AEDs = anti-epileptic drugs, AOR = adjusted odds ratio, BMI- body mass index, CI = confidence interval, COR = crude odds ratio, FEV = fruits and vegetable, LDL-c = low-density lipoprotein, N/A = Not Applicable, TC = total cholesterol.

NCEP-ATP III criteria for MS, the multivariable regression analysis showed that taking multiple antiepileptic agents (AOR = 6.81, 95% CI: 1.29–35.92), having TC ≥ 200 mg/dl (AOR = 7.37, 95% CI: 1.32–41.13) and BMI (AOR = 1.53, 96% CI: 1.16–2.01) were found to have significant associations with MS (Table 7).

## Discussion

Metabolic syndrome is a group of metabolic risk factors including glucose intolerance, dyslipidemia and hypertension which are associated with an increased risk of T2DM and CVDs [4]. In view of the fact that information from diverse studies demonstrates the risk of MS-associated development of CVDs is common for individuals with epilepsy, this study aimed to determine the magnitude and associated factors of MS among epileptic patients attending DCSH and compare it with the respective healthy control groups. It has been revealed the hidden MS among epileptic patients.

The study found that the prevalence rate of MS in the epilepsy group was 25.5% (95% CI: 17.03%- 33.95%) as per NCEP ATP- III criteria and 23.5% (95% CI: 15.28%- 31.76%) as per IDF criteria. This finding is in line with similar studies conducted in Kigali Rwanda using ATP

**Table 7. Bivariable and multivariable binary logistic regression analysis of factors associated with metabolic syndrome among the epileptic group using NCEP-ATP III criteria at DCSH, Dessie, Northeast Ethiopia, 2021 (n = 102).**

| Variables | Categories | MS- NCEP-ATPIII | | COR (95% CI) | AOR (95% CI) | p-value |
|---|---|---|---|---|---|---|
| | | Yes (n (%) | No (n (%)) | | | |
| Age category | 18–28 | 6 (17.6) | 28 (82.4) | 1 | 1 | |
| | 29–39 | 9 (25.0) | 27 (75.0) | 1.56 (0.49–4.96) | 1.62 (0.27–9.43) | 0.598 |
| | ≥40 | 11 (34.4) | 21 (65.6) | 2.44 (0.78–7.67) | 1.92 (0.28–13.19) | 0.509 |
| Residence | Rural | 13 (34.2) | 25 (65.8) | 1 | 1 | |
| | Urban | 13 (20.3) | 51 (79.7) | 0.49 (0.19–1.21) | 0.54 (0.12–2.48) | 0.432 |
| Level of physical activity | Sufficient | 11 (16.9) | 54 (83.1) | 1 | 1 | |
| | Low | 15 (40.5) | 22 (59.5) | 3.35 (1.33–8.42) | 3.04 (0.76–12.17) | 0.116 |
| FEV-intakes | Adequate | 7 (15.6) | 38 (84.4) | 1 | 1 | |
| | Low | 19 (33.3) | 38 (66.7) | 2.71 (1.02–7.21) | 2.70 (0.64–11.40) | 0.176 |
| Epilepsy subtype | Generalized onset | 19 (29.2) | 46 (70.8) | 1 | 1 | |
| | Focal onset | 1 (6.7 | 14 (93.3) | 0.17 (0.02–1.41) | 0.39 (0.03–6.04) | 0.505 |
| | Unknown Onset | 6 (27.3) | 16 (72.7) | 0.91 (0.31–2.67) | 0.75 (0.15–4.27) | 0.792 |
| Epilepsy dura-tion (mean ±SD) | N/A | 7.6 ± 4.8 | 5.4 ± 3.1 | 1.17 (1.04–1.32) | 1.01 (0.83–1.23) | 0.920 |
| Current AEDs use | On monotherapy | 13 (19.1) | 55 (80.9) | 1 | 1 | |
| | On Poly therapy | 12 (52.2) | 11 (47.8) | 4.62 (1.67–12.16) | 6.81 (1.29–35.92) | *0.024 |
| | Not on AEDs | 1 (9.1) | 10 (90.9) | 0.42 (0.05–3.6) | 1.23 (0.09–16.44) | 0.878 |
| Drug responsiveness status | Drug responsive | 20 (31.7) | 43 (68.3) | 1 | 1 | |
| | Drug-resistant | 2 (20.0) | 8 (80.0) | 0.54 (0.11–2.76) | 0.11 (0.01–1.72) | 0.115 |
| | Undefined | 4 (13.8) | 25 (86.2) | 0.34 (0.11–1.12) | 0.20 (0.03–1.27) | 0.088 |
| BMI (mean ±SD) | N/A | 24.73± 2.9 | 21.7± 2.7 | 1.48 (1.22–1.80) | 1.53 (1.16–2.01) | *0.002 |
| TC | <200 | 13 (17.6) | 61 (82.4) | 1 | 1 | |
| | ≥ 200 | 13 (46.4) | 15 (53.6) | 4.07 (1.57–10.56) | 7.37 (1.32–41.13) | *0.023 |
| LDL-C | <130 | 12 (16.2) | 62 (83.8) | 1 | 1 | |
| | ≥ 130 | 14 (50.0) | 14 (50.0) | 5.17 (1.97–13.56) | 1.58 (0.36–6.99) | 0.544 |

**Note**: Sufficient physical activity was considered for participants having either moderate or vigorous levels of activity (according to WHO definition), 1-References, *Statistically significant factors associated with MS.

**Abbreviations**: AEDs = anti-epileptic drugs, AOR = adjusted odds ratio, BMI- body mass index, CI = confidence interval, COR = crude odds ratio, FEV = fruits and vegetable, LDL-c = low-density lipoprotein, N/A = Not Applicable, TC = total cholesterol.

III (30.6%) [7], in Estonia using ATP-III (20.3%) [22], in India by ATP III criteria (29.5%) [23] and in Istanbul Turkey using IDF criteria (32.6%) [24]. Conversely, a higher rate of MS as compared to the finding of this study was reported among epileptic patients, 52.6% in South India using the AHA/NHLBI [8], 47.2% based on IDF Criteria, and 39.3% based on ATP-III criteria in Brazil [25], 43.5% in Italy using ATP-III [26] and 47.2% in West China based on AHA/NHLBI criteria [27]. The possible reason for this discrepancy may be the difference in sample size and sampling technique, the differences in study approaches (different in patient's selection criteria such as age, weight and anticonvulsants medications), the difference in types of anticonvulsant agents utilized by the patients and the difference in socio-economic status. For instance, in West China and Italy, the sample size was 36 patients taking VPA and 46 patients who were on monotherapy respectively, while in this study the sample size was 102 and consider all epileptic patients regardless of the medication status. Moreover, in Brazil, the participants were taking antiepileptic drugs (AEDs) for at least one year, and at least two years in South India, but in this study, all epileptic patients taking AEDs (regardless of the duration), and not taking AEDs were included. In addition to the above reasoning, the discrepancy may

be partly attributed to the various criteria employed by the studies. For example, in West China and South India, the diagnostic criteria were AHA/NHLBI. Other previous studies also reported a higher prevalence of MS: 47% among Chinese obese patients with epilepsy on VPA [27] and 43.5% in Italian overweight epileptic patients treated with VPA [26] according to NCEP ATP III and IDF definitions, respectively. A lower prevalence of MS as compared to the current study was also reported in epileptic patients in India (14.7%) and Iraq (5.7%) [28] using IDF and NCEP-ATPII criteria respectively. The possible reason for this discrepancy might also be the difference in patients' selection criteria, the types of anticonvulsant agents utilized by the patients and the difference in socio-economic status.

On the other hand, the study observed a 13.7% (95% CI: 7.04%- 20.40%) and 14.7% (95% CI: 7.83% - 21.58%) prevalence of MS among the control groups using NCEP ATP- III and IDF criteria, respectively. This finding is consistent with the study conducted in West Gojjam (17.3%) [14], in Mizan-Aman town (9.6%) [29], according to the modified NCEP-ATP III criteria, and in Eastern Ethiopia (20.1%) [30] and Nigeria (18%) [31] according to IDF criteria. However, it is low as compared with the pooled review among the Ethiopian population (27.92%) [13], a study from Ghana (35.9%) [32], and India (33.5%) [33] using IDF criteria. The discrepancy could be due to differences in sample size, study setting, and study population.

In comparing the prevalence of MS between the two groups, the study observed a higher prevalence rate in the epilepsy group (25.5%) compared to the healthy controls (13.7%) as per NCEP ATP- III criteria with a statistically significant difference. Similarly, a higher prevalence rate of MS was estimated in epilepsy (23.5%) as compared to the control group (14.7%) using IDF criteria, but not statistically significant. Even though there are limitations in comparing our data with the previously similar published studies where the MS in epileptic patients was studied in non-comparative studies, few studies report a consistent result with our finding: a study conducted in Istanbul, Turkey (32.6% for epilepsy $v_s$ 12.0% for healthy controls) [24] and China (47.2% for epilepsy $v_s$. 20.1% for controls) based on IDF criteria [27].

The higher prevalence in epileptic patients could be understood from the perspective of seizure-related metabolic abnormalities, long-term antiepileptic medications use, and a more sedentary lifestyle due to epilepsy [8–10,22,26]. It has been hypothesized that epileptic seizures damage specific brain nuclei in the hypothalamus and can change serum levels of some neurotransmitters and hormones, which leads to an imbalance of food intake and energy expenditure with subsequent weight gain [34–36]. Moreover, the majority of epileptic participants in this study (89.1%) were on different anti-epileptic agents, which often lead to weight gain, dyslipidemia as well as increase the risk of metabolic disturbances and MS by themselves [9,22,37,38]. It could also be explained by the high occurrence of a sedentary lifestyle in PWE. People with epilepsy tend to participate less often in physical activities, due to fear of seizures (concerns of injury) and social embarrassment compared to subjects without epilepsy [39]. This study also supports the above studies and found a significantly higher prevalence of low physical activity in epileptic participants (36.6%) than the healthy controls (21.6%). Other mechanisms possibly contributing to higher rates of MS in the group of patients with epilepsy could be higher activation of stress pathway through the hypothalamic-pituitary-adrenal axis although these have not been investigated independently [40–42]. This leads to overactivity of the sympathoadrenal system with the release of counter-regulatory hormones in a chronic state and finally predisposes to dyslipidemia as well as insulin resistance. Furthermore, the observed significant difference in the prevalence of some components of MS between epilepsy and control groups also suggests the difference in metabolic disturbances between the groups and MS in isolation.

In contrast to our finding, a comparative cross-sectional study conducted in Estonia reported a relatively lower magnitude of MS in the epilepsy group than the healthy controls, 20.3% V$_s$ 29.9% using ATP-III criteria [43]. However, the proportion reported to both groups was inconsistent and was below or above the reported levels of the present study. The discrepancy could be due to the age variation of the study populations or the difference in drug use. For instance, in the Estonian study population, most of the participants in the control group were older than the epileptic groups (median ages: 32 and 47 years respectively). It has been reported that age has a direct relation with MS as aging is related to physical inactivity and physiological changes like increased fat mass (abdominal fat deposition) accompanied by decreased muscle mass, hence concomitant insulin resistance.

Among all MS components analyzed, the most common in both groups was reduced high-density lipoprotein cholesterol (HDL-C) at 40.6% in epileptic participants and 26.7% in controls, with a statistically significant difference. The higher rate of reduced HDL-C in epileptic patients can be explained by the use of AEDs and the high rate of abdominal obesity than the control (27.7% Vs 19.8%, using IDF criteria), although the difference was not statistically significant. On top of that, significantly higher prevalence rates of some other MS components such as increased TG (33.7% Vs 20.8%) and raised FBG (15.8% v$_s$ 4.0% by ATP-III) were observed in epilepsy compared to the controls thereby suggesting that the metabolic disturbances are different between the two groups. Hyperglycemia may be due to drugs that are known to cause hyperglycemia like VPA or due to a combination of MS risk factors commonly found among patients with epilepsy like obesity and lower exercise capacity that make body cells less sensitive or resistant to insulin.

Regarding associated factors of MS in epilepsy, a significant association of some variables with MS was observed using both ATP-III and IDF diagnostic criteria. According to IDF criteria, a low level of physical activity was significantly associated with MS, and participants with low physical activity levels were more likely to have MS compared to their counterparts. Evidence studied in Spain [12] and India [9] showed that a low level of physical activity was positively associated with MS compared to sufficient activity. This might be due to the effect of sufficient physical exercise on burning more energy which can prevent the accumulation of fat and weight gain. Besides, exercise can improve insulin sensitivity and build muscle mass rather than fat mass, hence having a protective role on metabolic risks [44]. Therefore, having insufficient physical activity increases the risk of MS.

Likewise, BMI was significantly associated with MS in both the IDF and ATP-III criteria. This is consistent with the studies conducted in Estonia [22], Iran [45] and Japan [38]. The result could be explained in light of the contribution of increased body weight to central obesity, which leads to the accumulation of fat in the body. Fat forms artery plaque, which narrows arteries and capillaries leading to hypertension (a component of MS) and decreasing insulin sensitivity, consequently leading to a greater risk of MS [46].

Elevated TC was also found to be significantly associated with a higher risk of having MS compared to normal TC levels, using both criteria. The possible justification is that elevated total cholesterol has a direct correlation with most MS components and an inverse relation with HDL-c. Increased cholesterol has been shown to increase increment in weight and visceral adiposity, which intern leads to an increased release of free fatty acids. This causes a decrease in insulin action and sensitivity. Insulin resistance prevents the use of glucose by causing protein kinase inhibition in the muscle but stimulates lipogenesis and gluconeogenesis by activating protein kinase in the liver. So, the blood glucose level rises. On the other hand, it also facilitates the formation of hypertension by causing vasoconstriction. Thus, MS development may be easier for a high level of TC.

Moreover, epileptic participants in this study taking multiple anticonvulsant medications were found to have a significantly higher risk of developing MS in both the IDF and ATP-III criteria, which corroborates with other previous studies conducted in India [9,47] and Estonia [22]. This could be explained by the different effects of AEDs on weight subsequently affecting the development of insulin resistance and MS. Enzyme inducers AEDs (CPZ, phenytoin, phenobarbitone) have a direct effect on lipid metabolism, which enhances hepatic P450 cytochrome system activity (that involves in the synthesis of serum cholesterol) leading to increased cholesterol synthesis [48,49]. In addition, some AEDs may induce direct stimulation of the hypothalamus via the GABA pathway (leading to deregulation of the neuroendocrine control of energy intake) and alterations of adipokine gene expression in the brain and pituitary that alter adipokines released from adipose tissue- causing appetite stimulation [50]. It can finally promote hyperleptinemia leading to leptin resistance and over-secretion of insulin resulting in insulin resistance, hence can increase the risk of having MS [50].

## Strength and limitation of the study

As a strength, this study was the first that attempted to determine the prevalence of MS and associated factors among epilepsy patients in Ethiopia, hence ultimately adding to the limited data. It also includes healthy controls as a comparative group and treatment-naive epileptic patients. Despite these strengths, the study has the following limitations. First, our study design was cross-sectional by its nature and signifying that it cannot sufficiently ascertain the causal association between MS and associated factors. Second, a small sample size (due to budget and time constraints) and the sampling coming from one Referral Hospital at a specific period, may limit external generalizability. Third, as the study was conducted in a referral care center, the group of epilepsy patients selected may reflect more resistant cases of epilepsy than those in the general population. Moreover, only two definitions were used to assess the prevalence of MS; a different prevalence rate could have been observed if other MS definitions like the WHO definition were used.

## Conclusion

In general, our findings indicated that people with epilepsy are more prone to develop MS and its components than healthy controls using ATP-III criteria, indicating that MS is a problem for epileptic patients. Hence, they are at increased risk of developing complications such as CVDs and premature mortality. Among the components of MS, HDL-c was the most commonly encountered abnormality, followed by elevated TG levels in both groups. Moreover, low physical activity, BMI, taking multiple AEDs and having raised TC were significantly associated with MS in the epileptic group.

## Supporting information

**S1 Fig. Analysis on the authenticity of IDF criteria referring to the NCEP-ATP III criteria.** (DOCX)

**S1 File. WHO STEPS questioner.** (DOCX)

**S1 Dataset. Datasets used for the analysis.** (DTA)

## Acknowledgments

Declaration

We would like to give our thanks to the University of Gondar for supporting us to do this work. We also owe our heartfelt gratitude to Dessie Comprehensive Specialized Hospital for collaborating on the study and data collectors and study participants who were facilitating and supporting us during this work.

## Author Contributions

**Conceptualization:** Altaseb Beyene Kassaw, Hiwot Tezera Endale, Kibur Hunie Tesfa, Meseret Derbew Molla.

**Data curation:** Altaseb Beyene Kassaw, Kibur Hunie Tesfa, Meseret Derbew Molla.

**Formal analysis:** Altaseb Beyene Kassaw, Hiwot Tezera Endale, Meseret Derbew Molla.

**Investigation:** Altaseb Beyene Kassaw, Meseret Derbew Molla.

**Methodology:** Altaseb Beyene Kassaw, Hiwot Tezera Endale, Kibur Hunie Tesfa, Meseret Derbew Molla.

**Software:** Altaseb Beyene Kassaw, Hiwot Tezera Endale, Kibur Hunie Tesfa.

**Supervision:** Kibur Hunie Tesfa.

**Validation:** Altaseb Beyene Kassaw, Hiwot Tezera Endale, Kibur Hunie Tesfa, Meseret Derbew Molla.

**Visualization:** Meseret Derbew Molla.

**Writing – original draft:** Altaseb Beyene Kassaw, Hiwot Tezera Endale, Kibur Hunie Tesfa, Meseret Derbew Molla.

**Writing – review & editing:** Altaseb Beyene Kassaw, Hiwot Tezera Endale, Kibur Hunie Tesfa, Meseret Derbew Molla.

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
