## [Decision Letter · Decision Letter 0]

24 Aug 2022

PONE-D-22-08200Metabolic syndrome and its associated factors among epileptic patients at Dessie Comprehensive Specialized Hospital, Northeast Ethiopia; a hospital-based comparative cross-sectional studyPLOS ONE

Dear Dr. kassaw,

Thank you for submitting your manuscript to PLOS ONE. After careful consideration, we feel that it has merit but does not fully meet PLOS ONE’s publication criteria as it currently stands. Therefore, we invite you to submit a revised version of the manuscript that addresses the points raised during the review process.

We look forward to receiving your revised manuscript.

Kind regards,

Rick J. Jansen, PhD, MS

Academic Editor

PLOS ONE

https://journals.plos.org/plosone/s/file?id=ba62/PLOSOne_formatting_sample_title_authors_affiliations.pdf".

“there has been no funding for this work that could have influenced its outcome”

“No authors have competing interests”

Important: If there are ethical or legal restrictions to sharing your data publicly, please explain these restrictions in detail. Please see our guidelines for more information on what we consider unacceptable restrictions to publicly sharing data: http://journals.plos.org/plosone/s/data-availability#loc

unacceptable-data-access-restrictions. Note that it is not acceptable for the authors to be the sole named individuals responsible for ensuring data access.

5. PLOS requires an ORCID iD for the corresponding author in Editorial Manager on papers submitted after December 6th, 2016. Please ensure that you have an ORCID iD and that it is validated in Editorial Manager. To do this, go to ‘Update my Information’ (in the upper left-hand corner of the main menu), and click on the Fetch/Validate link next to the ORCID field. This will take you to the ORCID site and allow you to create a new iD or authenticate a pre-existing iD in Editorial Manager. Please see the following video for instructions on linking an ORCID iD to your Editorial Manager account: https://www.youtube.com/watch?v=_xcclfuvtxQ.

Additional Editor Comments:

Please make sure to address all reviewer comments fully with specific attention to the comment on reproduction of existing work. Thank you!

Reviewers' comments:

Reviewer's Responses to Questions

**Comments to the Author**

1. Is the manuscript technically sound, and do the data support the conclusions?

Reviewer #1: Partly

Reviewer #2: Yes

Reviewer #3: No

2. Has the statistical analysis been performed appropriately and rigorously? 

Reviewer #1: Yes

Reviewer #2: Yes

Reviewer #3: No

3. Have the authors made all data underlying the findings in their manuscript fully available?

Reviewer #1: Yes

Reviewer #2: Yes

Reviewer #3: No

4. Is the manuscript presented in an intelligible fashion and written in standard English?

Reviewer #1: Yes

Reviewer #2: No

Reviewer #3: No

5. Review Comments to the Author

Reviewer #1: in methology part the selection of apparently healthy individuals is not clear. so it needs revision

the selection of apparaently healhty individual criteria must be written on the material and method part.

Reviewer #2: Abstract: page 2 line 33, 36 & 39: the association of BMI with MS needs to be described clearly (higher vs lower BMI)

Introduction: paragraph one and two can be merged and re-written considering the paper is meant for the scientific community with an understanding of what epilepsy and metabolic syndrome are. The emphasis in the introduction should be on the cross-link between epilepsy and metabolic syndrome.

Study participants: Line 110-121: The authors would benefit in adding statement explaining why they chose to use healthy controls.

Result and discussion: why did the author prefer to use both the IDF and NCEP-ATP III criteria in the description of the result and discussion? Which one is used in the study setting? The authors should use one of the criteria and present the other as a sensitivity analysis by providing the result as a supplementary file.

Discussion: The authors frequently used statements mentioned in the result part in their discussion and focused on comparing their result with others. Instead of comparing the finding of your study with other studies, the focus should be on explaining the consequence of the study finding on the study population.

Strength and Limitation of the Study: Line 501 “It also includes healthy controls as a comparative group and treatment-naive epileptic patients.”….did the study include treatment naïve epileptic patients as a control group. As this is not reflected in neither the methods nor result section.

Reviewer #3: I have reviewed with great interest the original research article entitled "Metabolic syndrome and

its associated factors among epileptic patients at Dessie Comprehensive Specialized Hospital,

Northeast Ethiopia; a hospital-based comparative cross-sectional study" which was intended to

assess the magnitude of metabolic syndrome among epileptic patients. However, I have the

following concerns:

A sample size of 102 may not be enough to say the prevalence/magnitude of metabolic syndrome

among epileptic patients; rather, it might provide evidence for the presence of metabolic

syndrome in epileptic patients in comparison with another 102 healthy controls simply.

However, whether the metabolic syndrome is because of the anti-epileptic drugs or due to the

epilepsy disease per se should be clearly discussed in this manuscript.

Using the nonstandard abbreviations in the abstract section of the manuscript is not

recommended. However, the authors used several abbreviations in the abstract section, and the

author should write the long form of abbreviations for the first time, e.g., DCSH, MS, AEDs.

Since they did not have a list of healthy controls, how could they use a systematic random

sampling technique to select patient attendants or caregivers? The author said that they used

systematic random sampling techniques to select the control groups, which is impractical without

having the frame.

The sampling technique and procedures are not well explained for cases and control groups

separately. They should review the findings in relation to the sampling method.

Some of the major concerns are

1. Data do not supports the conclusions.

2. statistical analysis has been not performed appropriately and rigorously. Apparently it seems forged data.

3. Language is not clear, correct and there are ambiguous sentences as well. Besides, there are some dead sentences are there in the manuscript.

4. The work presented in the manuscript is simply copy of previously published work at another settings.

6. PLOS authors have the option to publish the peer review history of their article (what does this mean?). If published, this will include your full peer review and any attached files.

Reviewer #1: No

Reviewer #2: No

Reviewer #3: No

---

## [Author Response · Author response to Decision Letter 0]

28 Sep 2022

Response to the editor: We thank you for the insightful comments given to our manuscript which obviously further improves the quality of our work. We have been able to incorporate changes to reflect all the comments and suggestions provided.

Being specific on the points you raised:

#1. We will deposit soon our laboratory protocols in protocols.io., as it has been recommended

#2. Regarding financial disclosure, we have not made a change (we didn’t receive any funding for this study). Hence, we have stated “The authors received no specific funding for this work” on the cover letter.

#3. On the competing interest section, we have completed the Competing Interests on the online submission form to state any Competing Interests and have stated on the cover letter as “The authors have declared that no competing interests exist”.

#4. In Data Availability statement, we have specified the minimal data set underlying the results described in our manuscript (as Supporting information files). 

#5. ORCID iD for the corresponding author has been linked to Editorial Manager profile.

#6. We incorporate the ethics statement in the methods section of our manuscript. 

#7. We have fully addressed all the reviewer comments and suggestions.

Reviewer #1:

Response to the comments: Thanks for your kind reminders. We found your comment helpful and has now been written clearly as suggested. An equal number of age and sex-matched apparently healthy volunteer subjects (apparently healthy care givers/ patient attendants who full fill the inclusion criteria) were enrolled in the comparison group. The participants were selected consecutively. [ Page-6, L127-129], on unmarked version of our revised manuscript. 

Reviewer #2:

Response to Comment 1: thank you very much for your suggestion. Although we agree that categorizing BMI as high or low is good to assess its association with MS, continuous independent variables can also be fitted as such and would have high statical power in predicting association, strong prediction contribution to the outcome variable (Jill C. Stoltzfus, 2011). Accordingly, we take BMI as continuous variable and asses its association with MS. The clarified result has presented on “Associated factors of metabolic syndrome among the epileptic groups” section [ Page-20, L341-346, or on Table 7(Page 21) and Table 8 Page 22)]. 

Response to Comment 2: Thanks for your kind reminders. We found your comment is helpful and have revised accordingly.

Response to comment 3: We thanks for the suggestion and incorporated it in our revised manuscript [Page-5, L107-109]. To be clearer, we initially intended to show “whether MS is a problem in epileptic population in the study setting, or to assess whether MS is associated with epilepsy or weather epilepsy and related issues are a risk for MS”. Accordingly, we tried to compare the presence of MS between epileptic and non-epileptic study participants even though the small sample size we enrolled and the study design we used may not sufficiently ascertain. To minimize the effect of confounder, we chose non-epileptic group from healthy individuals. 

Response comment 4: we are grateful for the comment! Using a single criterion may over estimate or under estimate the magnitude of MS. Hence, we used both the IDF and NCEP-ATP III criteria independently in our setting. Moreover, we did sensitivity analysis and provided the result as a supplementary file (S1).

Response to comment 5: We thank you for you’re a very interesting comment. As the Purpose of discussion is to describe, analyzes and interpret/justify findings in relation to the existing literature, theory and practice ((Skelton.J. et al, 2000), (Shona Mc Combes, 2022), (Oner Şanli, et al, 2013)) we were trying to do accordingly. We believe the “consequence of the study finding on the study population” is addressed in a conclusion section [Page-29, Line no. 512-518].

Response to comment 6: Thanks for your kind question. We rather considered current treatment status of epileptic participants as a variable and fitted into logistic regression analysis (Current anti-epileptic drugs (AEDs) use: On mono therapy, On Poly therapy, Not on antiepileptic-agents/ treatment naïve), and tried to show weather the MS is because of the epilepsy by itself or because of the different anti-epileptic drugs [Table 7(Page 21) and Table 8 Page 22)]. 

Reviewer #3:

Response to the comment 1: We are very appreciative for your insightful suggestions and comments. We have gone through your comments carefully and tried our best to address them. Even though the study design we choice (cross-sectional study design) cannot sufficiently ascertain whether the metabolic syndrome is because of the anti-epileptic drugs or due to the epilepsy disease per se (which has been stated as a limitation), the likelihood of MS, given its criteria, has been adjusted for possible confounding variables using a multivariate logistic regression model [Table 7(Page 21) and Table 8 (Page 22)]. Hence, Current anti-epileptic drugs (AEDs) use status (On mono therapy, On Poly therapy, Not on antiepileptic-agents/ treatment naïve) has been adjusted to….

Response to the comment 2: thank you for pointing this out. We agree with this comment and has now been amended in the revised abstract.

Response to the comment 3: We thanks for a very interesting doubt you raised. Honestly speaking, volunteer healthy patient attendants were selected consecutively and we are very sorry for a typing fault that has been done in editing of the manuscript. We have now added the corrected content to the manuscript on [Page-6, Line no. 127-129].

Response to the comment 4: Thank you for the comments. We have revised these points according to the suggestions.

---

## [Editor Report · Decision Letter 1]

27 Oct 2022

PONE-D-22-08200R1Metabolic syndrome and its associated factors among epileptic patients at Dessie Comprehensive Specialized Hospital, Northeast Ethiopia; a hospital-based comparative cross-sectional studyPLOS ONE

Dear Dr. kassaw,

Thank you for submitting your manuscript to PLOS ONE. After careful consideration, we feel that it has merit but does not fully meet PLOS ONE’s publication criteria as it currently stands. Therefore, we invite you to submit a revised version of the manuscript that addresses the points raised during the review process. Please make sure to remove all nonstandard abbreviations form the abstract. Also, please address the comment from the reviewer suggesting this is a mostly reproduced manuscript of a published manuscript. Thank you!

We look forward to receiving your revised manuscript.

Kind regards,

Rick J. Jansen, PhD, MS

Academic Editor

PLOS ONE

Journal Requirements:

Additional Editor Comments:

Please make sure to remove all nonstandard abbreviations form the abstract. Also, please address the comment from the reviewer suggesting this is a mostly reproduced manuscript of a published manuscript. Thank you!
---

## [Author Response · Author response to Decision Letter 1]

21 Nov 2022

Dear Editor,

I would like to say thank you for your comments, support, and invitation for resubmitting our manuscript

kind regards.

Altaseb Beyene Kassaw

---

## [Editor Report · Decision Letter 2]

12 Dec 2022

Metabolic syndrome and its associated factors among epileptic patients at Dessie Comprehensive Specialized Hospital, Northeast Ethiopia; a hospital-based comparative cross-sectional study

PONE-D-22-08200R2

Dear Dr. kassaw,

We’re pleased to inform you that your manuscript has been judged scientifically suitable for publication and will be formally accepted for publication once it meets all outstanding technical requirements.

Kind regards,

Rick J. Jansen, PhD, MS

Academic Editor

PLOS ONE

Additional Editor Comments (optional):

The comments have been addressed adequately.
---

## [Editor Report · Acceptance letter]

19 Dec 2022

PONE-D-22-08200R2 

Metabolic syndrome and its associated factors among epileptic patients at Dessie Comprehensive Specialized Hospital, Northeast Ethiopia; a hospital-based comparative cross-sectional study 

Dear Dr. Beyene Kassaw:

I'm pleased to inform you that your manuscript has been deemed suitable for publication in PLOS ONE. Congratulations! Your manuscript is now with our production department. 

Kind regards, 

on behalf of

Dr. Rick J. Jansen 

Academic Editor

PLOS ONE